



# A micro-genetic algorithm for combinatorial optimization of physics parameterizations in Weather Research and Forecasting model for quantitative precipitation forecast in Korea

Sojung Park[1], Seon K. Park[1, 2, 3, 4]

[1]Department of Climate and Energy Systems Engineering, Ewha Womans University, Seoul, 03760, Korea
[2]Department of Environmental Science and Engineering, Ewha Womans University, Seoul, 03760, Korea
[3]Center for Climate/Environment Change Prediction Research, Ewha Womans University, Seoul, 03760, Korea
[4]Severe Storm Research Center, Ewha Womans University, Seoul, 03760, Korea

*Correspondence to*: Seon K. Park (spark@ewha.ac.kr)

**Abstract.** One of biggest uncertainties in Numerical Weather Predictions (NWPs) comes from treating the subgrid-scale physical processes. For the more accurate regional weather/climate prediction by improving physics parameterizations, it is important to optimize a combination of physics schemes as well as unknown parameters in NWP models. We have developed an interface system between micro-Genetic Algorithm ($\mu$-GA) and the WRF model for the combinatorial optimization of CUmulus (CU), MicroPhysics (MP), and Planetary Boundary Layer (PBL) schemes in terms of quantitative precipitation forecast for heavy rainfall events in Korea. The $\mu$-GA successfully improved simulated precipitation despite the non-linear relationship among the physics schemes. During the evolution process, MP schemes control grid-resolving scale precipitation while CU and PBL schemes determine subgrid-scale precipitation. This study has demonstrated the combinatorial optimization of physics schemes in the WRF model is one of possible solutions to enhance the forecast skill of precipitation.

## 1 Introduction

For numerical weather forecasting to be accurate, a numerical model should be able to represent real atmospheric conditions in terms of dynamics (i.e., governing equations), physics (i.e., parameterizations), and numerics (e.g., resolution and coordinate system). It should be also provided more accurate initial conditions. One of the biggest uncertainties in Numerical Weather Predictions (NWPs) comes from treating the subgrid-scale physical processes that have not been sufficiently understood. The subgrid-scale physical processes are parameterized in NWP models through empirical evidences, such as the derived value from observation and/or theoretical backgrounds. Therefore, the accuracy of physics parameterizations strongly depends on the followings: the value of parameters in given uncertainty ranges in parameterization schemes; the choice of parameterization schemes for each corresponding physical process. Note that prior to model simulation the unknown parameters and schemes should be fitted to the regional weather/climate to reduce considerable uncertainties in models.





The NWP models have several categories of subgrid-scale physical processes — shortwave and longwave radiation transfer, CUmulus (CU), MicroPhysics (MP), Planetary Boundary Layer (PBL), land surface processes, etc. In general, a model consists of a determined physics package, such as Unified Model (UM) (Cullen, 1993; Brown et al., 2012), Global Forecast System (GFS), and Korean Integrated Model (KIM) (Hong et al., 2018), whereas some models, such as the Weather

Research and Forecasting (WRF) model and the community Noah Land Surface Model (LSM) with Multi-Parameterization options (Noah-MP) (Niu et al., 2011), have each physical category with several optional parameterization schemes so that users can select the schemes.

The sensitivity experiments of the WRF model to physical parameterization schemes in simulating rainfall systems have been conducted (e.g., Crétat et al., 2012; Cohen et al., 2015; Evans et al., 2012; Song and Sohn, 2018). Cohen et al. (2015)

introduced the various PBL schemes employed by WRF model and examined sensitivity to PBL schemes in simulating cold season severe weather events occurred in the southeastern United States. Evans et al. (2012) explores the performance of various combinations of PBL, CU, MP, and radiation schemes for the rainfall events near East Coast Lows. They suggest that the Mellor-Yamada-Janjic PBL scheme and the Betts-Miller-Janjic CU scheme can be selected in this region with some robustness.

Previous studies on numerical weather and climate prediction model have manually examined one or several suitable schemes for rainfall events as a sensitivity test, hence not all schemes can be investigated, only selected combinations of schemes. This is because it takes a lot of computer resources and time to conduct the sensitivity test with all physics schemes. However, in addition to model development, all physics schemes need to be explored to simulate more accurate local weather and climate systems. Combinatorial optimization for a system or model configuration has been applied to the water

distribution system (Behzadian et al., 2009; Gupta et al., 1999; Simpson et al., 1994; Weng and Liaw, 2005), groundwater monitoring design (Babbar-Sebens and Minsker, 2010), and multi-reservoir operation (Chen et al., 2017). Many algorithms have been developed for combinatorial optimization such as simplex, integer programming, simulated annealing, and Genetic Algorithm (GA).

In weather and climate prediction models, most applications of GA have focused on the optimization of empirical parameters

in the model to fit the modeled properties (e.g., precipitation) to observed counterparts (e.g., Lee et al, 2006; Yu et al, 2013). However, in this study, we created a micro-GA-WRF ($\mu$-GA-WRF) interface to seek the optimal set of CU, MP, and PBL schemes in the WRF model for rainfall events in terms of quantitative precipitation forecast (QPF). The $\mu$-GA-WRF interface system is to automatically extract the optimal scheme combination from physics options in the WRF model. The present study attempts to find an optimal combination of parameterization schemes, which is a new and challenging task.

This paper is organized as following: Section 2 illustrates background of combinatorial optimization; Section 3 explains data and method, including the $\mu$-GA-WRF interface system; case study are presented in Section 4; Section 5 contains concluding remarks.



## 2 Background of combinatorial optimization

To select a suitable optimization algorithm, we should consider the characteristics of objective functions, control variables as
well as optimization problems. Jamil and Yang (2013) reviewed and compiled benchmark functions found from all the
available literature for global optimization problems. They focused on the diverse properties of objective functions such as
continuity, linearity, modality, separability, and dimensionality. In terms of combinatorial optimization, control variables can
be discretized and indexed values, and the value of the control variable itself can be meaningless. These discrete control
variables make the solution space of the cost function discontinuous. Therefore, it is important to choose an algorithm that
can handle these properties.

The GA as an evolutionary algorithm is based on the natural selection of genes (i.e., parameters in the algorithm) to search
for the optimum. Research has adopted the GA to solve network system design optimization problems with a growing trend
from the end of the 20th century (e.g., Simpson et al., 1994; Halhal et al., 1997; Savic and Walters, 1997; Pilar et al., 1999;
Dandy et al., 2001). Gupta et al. (1999) emphasized the GA has advantages of using discrete variables for optimization and
having an insensitive initial solution (i.e., robustness in the initial solution). Azadivar and Tompkins (1999) applied the GA
approach to optimize to qualitative variables (e.g., structural design) in a manufacturing system, as simulation optimization.
The GA coupled with a simulation-model generator searches for the different combinations of design configurations and
evaluates the simulations. Gupta et al. (1999) shows that the GA provided a lower cost design of water distribution networks
(e.g., pipe networks), compared to the non-linear programming technique. Weng and Liaw (2005) established a
combinatorial optimization model, called the Sewer System Optimization Model for Layout & Hydraulics, to optimize cost-
effective designs for an urban sewer system. The better alternate network layouts were created more productively by
applying the GA. Davis et al. (2019) optimized a malaria model with the GA by clustering locations based on the
relationships between malaria and environmental drivers (e.g., temperature, precipitation, and vegetation index). To predict
environmentally-driven malaria outbreaks across a heterogeneous region, the GA optimized the number of clusters and the
environmental predictors for the districts in each cluster in the malaria model.

Furthermore, the GA was applied for combinatorial optimization to the Noah-MultiParameterization (Noah-MP) Land
Surface Model (LSM), which can be coupled with numerical weather prediction model (e.g., WRF model) in Hong et al.
(2014). Noah-MP was augmented with multiple physics options for 10 different land surface processes such as phenology,
snow, and groundwater (Niu et al., 2011). Hong et al. (2014) performed scheme-based model optimizations in simulating
evapotranspiration and runoff (i.e., water balance) in Noah-MP over the Han River basin in South Korea. In addition, they
showed a potential advantage of the Noah-MP and GA coupled system to model diagnosis — the evolutionary process
provides information on sensitivity and interrelationship of physics schemes with regard to further model calibrations and
improvements. Hong et al. (2015) further evaluated the applicability of the coupling system of micro-GA (i.e., an efficient
version of GA; $\mu$-GA) and Noah-MP to larger and multiple regions in East Asia.





GA does not perform a random search of the extrema, but performs gradual search toward the extrema. However, it does not directly use gradient information of an objective function, but instead mimics the evolutionary method to quickly reach the global optima. The gradual search is based on a fact that the best individual stands for the nearest point to the optima. In the case of temperature as a physical quantity with the continuous nature (e.g., real number), an increase or decrease of temperature is meaningful. On the other hand, for physics schemes with discontinuous nature (e.g., integer), the option of

schemes as an index has no physical meaning. The best option of physics schemes in the model is not related to the nearest options. Thus, the random search is more appropriate to optimize each of the schemes, however, the use of the evolutionary algorithms is reasonable when looking for a combination of physics schemes. Note that the combinatorial optimization must consider a randomness to avoid falling to the local optimum. The $\mu$-GA conduct the global search through random number generator as well as crossover operator, hence, the $\mu$-GA is a very useful tool for combinatorial optimization.

## 3 Data and methods

### 3.1 Observation data

The observation data have strongly affected the verification results (Rossa et al., 2008). Merged gauge-radar precipitation has the greatest advantage of the spatially uniform information available. The composite precipitation data produced by the Korea Meteorological Administration (KMA) Radar-Automatic weather station Rain-rate (RAR) system (Suk et al., 2013)

using 11 radars was employed to optimize the combination of the physics schemes in the WRF model. The observational domain covers 1241 km ×1761 km in the Korean Peninsula, centered at 38 °N and 126 °E in the Lambert conformal conic projection. It has enough horizontal (i.e., 1 km) and temporal resolution (i.e., 10 min) to compare with the precipitation fields obtained by the high resolution model. The performance of RAR system was examined for 10 heavy rainfall cases selected during the summer of 2006 in Suk et al. (2013), obtaining a squared correlation coefficient ($R^2$) of 0.84 between RAR-

estimated rainfall and the observed daily accumulated rainfall from rain gauges.

For comparison with model output, RAR-estimated rainfall data were aggregated to a 5 km resolution grid. A downscaled grid box represents the average of 25 original grid boxes. For reasonable representativeness of samples, we take the average if more than 80 % of original grid boxes have meaningful values.

### 3.2 $\mu$-GA-WRF interface system

#### 3.2.1 $\mu$-GA

The GA developed by John Holland in the 1970s is a global optimization approach based on the Darwinian principle of natural selection: stronger individuals in a generation are more likely to produce offspring. The aim of the GA is to find the best individual with either a maximum or minimum fitness by means of a stochastic global search of the solution space, through the generations. The algorithm applies the crossover and mutation operators to avoid local maximum/minimum





solutions. The $\mu$-GA (Krishnakumar, 1989) is an improved version of GA with smaller population sizes (e.g., of 5) and simplifies a generation to generation evolution, hence efficiently reducing the computational resources. To simplify the algorithm, the mutate operator is not used, but the crossover operator is used to increase the diversity at a rate of 100 %. Furthermore, whenever inner loop convergence is achieved, the new population for the next generation consists of all new random individuals, except one elite individual. Thus, $\mu$-GA can avoid trapping into the local optimum.

The flowchart of the $\mu$-GA interfaced to the WRF model is provided in Fig. 1. The selection operator in the $\mu$-GA is tournament selection with a shuffling technique to choose random pairs for mating. Fitness function to evaluate each individual is of the utmost importance in GA and it should be designed taking into account the perspective of the optimization. If inner loop does not converge, selection is performed and all populations go through a crossover process, then one of populations is altered by the elite. Here, elite individual from the previous generation is saved as one of

populations in the current generation (i.e., the elitism). Since the crossover probability of 1.0 is used without the mutation operator in the $\mu$-GA, each individual quickly resembles the elite through generations (i.e., inner loop). In other words, the optimization within the inner loop has a feature of local search by exploring specifically the solution space around the elite. The $\mu$-GA decides that an inner loop converges upon an optimum when the different chromosomes between an elite individual and all the others are less than 5 % as binary bits. After the inner loop convergence, all individuals in the next

generation, except for one surviving elite of the parent generation, are regenerated using random number generator, thus widening the search space (i.e., outer loop; global search). As a criterion of the outer loop convergence to finalize the algorithm, we commonly set the maximum number of generations.

### 3.2.2 $\mu$-GA-WRF interface system

We created the $\mu$-GA-WRF interface system to seek the optimal physics set of CU, MP, and PBL schemes in the WRF

model for rainfall events in terms of QPF. The WRF model, one of mesoscale NWP systems, has been developed for atmospheric research and operational forecasting applications from the latter 1990s by a collaborative partnership of the National Center for Atmospheric Research (NCAR), the National Oceanic and Atmospheric Administration (represented by the National Centers for Environmental Prediction (NCEP) and the Earth System Research Laboratory), the U.S. Air Force, the Naval Research Laboratory, the University of Oklahoma, and the Federal Aviation Administration (FAA). The detail of

model configuration is addressed in Sect. 3.3.

Figure 1 shows the flowchart of the $\mu$-GA-WRF interface system. In the $\mu$-GA-WRF interface system, the $\mu$-GA controls WRF simulations in the process of 'Compile & Run WRF'. First, the $\mu$-GA randomly initializes individuals (i.e., combinations of physics schemes) in the first generation. Through the inner and outer loop, the $\mu$-GA evaluates the fitness value calculated by WRF results and RAR observation, and reruns the WRF models with new scheme combinations. Finally,

if both the inner loop and outer loop converge, the optimization process in the $\mu$-GA-WRF interface system is finished.

The $\mu$-GA is implemented as input parameters. We set population size of 5, meaning that each generation has 5 individuals (i.e., model simulations). The maximum value of generations is set to 100, typically used in GA experiment. The number of



parameters (groups of bits) of each individual for GA is 3, which is the number of schemes to be optimized. We used single-point crossover.

### 3.2.3 Fitness function

Fitness is the basis for evaluating the superiority among individuals consisting of combinations of chromosomes. Designing a fitness function in GA is critical for optimizing the model as intended. In this study, we are trying to improve the model simulation in terms of QPF. Thus, we used the Equitable Threat Score (ETS; Hamil, 1999) as the fitness function, also called objective function. The fitness is computed by sum of ETSs within each precipitation threshold:

$$\text{Fitness} = \begin{cases} ETS_i, & for\ detection, i = 0.1\ or\ 3. \\ \sum_i ETS_i, & for\ heavy\ rainfall,\ i = 10, 20, 30, \cdots, 300. \end{cases} \quad (1)$$

Where $i$ are specified thresholds of accumulated precipitation in mm and the ETS and chance is defined as:

$$ETS = \frac{hits - chance}{hits + misses + false\ alarms - chance}, \quad (2)$$

$$chance = \frac{(hits + misses)(hits + false\ alarms)}{(hits + misses + correct\ negatives + false\ alarms)}. \quad (3)$$

ETS have values in the range from -1/3 to 1. The closer the ETS is to a unity, the better the forecast skill. While if ETS is equal to or less than 0, the forecast skill is the same as, or even worse than that of a random forecast. Here, hits, misses, false alarms and correct negatives from a 2×2 contingency table are estimated by the joint distribution of binary (yes/no) forecasts and observations (Table 1). Rainfall estimations can be evaluated through the table that explicitly provides prediction capability and types of errors in the prediction.

When focusing on heavy rainfall, total fitness value is calculated by the sum of ETS at the threshold ranging from 10 to 300 mm with an interval of 10 mm, whereas when focusing on the precipitation detection, a precipitation threshold of 3 mm is used. In order to detect precipitation, the threshold of precipitation accumulated over 24 or 12 hours is generally used as a value between 0.1 and 0.3 mm (Rossa et al., 2008). Park et al. (2018) obtained the threshold value of 3 mm h⁻¹ for the station average precipitation rate when the cumulative percentage of warm-season precipitation events in Korea reached approximately 80 % based on AWS observation data. As we would like to improve the forecast accuracy of precipitation in Korea, we selected the threshold value of 3 mm for hourly precipitation for the calculation of ETS. In this study, we also conducted the sensitivity test of precipitation accumulation period.

### 3.3 Experimental design

The WRF model (version 4.0.3) was initialized at 0000 UTC 5 August 2018 with the 6-hourly initial and boundary conditions given by the National Center for Environmental Prediction (NCEP) Final (FNL) Operational Model Global Tropospheric Analyses data on 1°×1° grids. The WRF model configuration is based on the followings: horizontal grid spacings of 25 and 5 km for two nested domains (Fig. 2); horizontal grid points of 60×60 and 116×136; the model top of 50 hPa with 33 vertical levels; Dudhia shortwave radiation scheme (Dudhia, 1989), Rapid Radiative Transfer Model (RRTM)





longwave radiation scheme (Mlawer et al., 1997), revised fifth generation National Center for Atmospheric Research (NCAR)/Penn State Mesoscale Model (MM5) surface layer scheme (Jimenez et al., 2012), and Unified Noah LSM (Chen et al., 1996; Koren et al., 1999). The control experiment referred to as CTL is simulated with Kain-Fritsch scheme (KF), the WRF Double-Moment (WDM) 6-class scheme, and the YonSei University (YSU) scheme as CU, MP, and PBL scheme, respectively, generally used to simulate precipitation system in Korea. The optimization results from the $\mu$-GA-WRF interface system are referred to as OTP.

We selected CU, MP, and PBL physical processes for the combinatorial optimization. The CU parameterization determines the prediction of sub-grid scale processes associated convective clouds and precipitation at a coarse resolution. Meanwhile, the MP regulates the gird-resolving processes of clouds. The PBL scheme, which could indirectly influence precipitation by interacting with other physics, can affect temperature and moisture profiles in the lower troposphere via exchanges of moisture, heat and momentum through the mixing associated with turbulent eddies. The options of CU, MP, and PBL schemes used for the optimization in the $\mu$-GA-WRF interface system are shown in Table 2. If Mellor-Yamada-Janjic (MYJ), Quasi-Normal Scale Elimination (QNSE), Mellor-Yamada Nakanishi and Niino Level (MYNN) 3, or Total Energy Mass Flux (TEMF) is selected as the PBL scheme, Eta similarity (Monin and Obukhov, 1954; Janjic, 1994, 1996, 2002), QNSE, MYNN, or TEMF should be set as a surface-layer scheme, respectively. The surface-layer scheme is the lowest part of the atmospheric boundary layer where the surface fluxes (i.e., surface heat, moisture, and momentum fluxes) can be calculated not only by combining the land-surface model, but also by itself. We have found the best scheme combination by the $\mu$-GA as the mechanical and objective optimization method without a model simulations of 2,688 (= 14×16×12) which is the total possible number of scheme combinations.

During the 12-hr period from 1200 UTC 5 to 0000 UTC 6 August 2018, precipitation was evaluated by fitness functions. We perform the optimization experiments with 5 different fitness functions based on ETS in Sec. 3.2.2. Table 3 shows the summary of experiments — OPT-EXP1 for 12 hourly accumulated precipitation with precipitation thresholds ranged from 10 to 300 mm with the interval of 10 mm; OPT-EXP2 for 12 hourly accumulated precipitation with precipitation threshold of 3 mm; OPT-EXP3 for all 6 hourly accumulated precipitation during the evaluation period with precipitation threshold of 3 mm;. OPT-EXP4 for all 3 hourly accumulated precipitation during the evaluation period with precipitation threshold of 3 mm; OPT-EXP5 for all hourly accumulated precipitation during the evaluation period with precipitation threshold of 3 mm. For OPT-EXP1, the total fitness value is calculated as the sum of the ETSs at all thresholds, while for OPT-EXP2 – OPT-EXP5, it is calculated as the average of the ETSs for each accumulated time over 12 hours.

## 4 Case study

### 4.1 Case description

A coastal flood occurred in Korea due to a quasi-stationary mesoscale convective system (MCS) which produced heavy rainfall on 5–6 August 2018. As the unexpected event, the back-building MCS was located in Yeongdong region about a day,



thus the heavy rainfall caused damage to property estimated at 177 million won (KMA, 2018). For the period from 1100 UTC 5 to 1400 UTC 6 August, 294.5 mm of precipitation was recorded at Sokcho. To predict more accurately, forecasters essentially need the mesoscale information from NWPs as well as synoptic weather charts, vertical soundings, satellite observations, weather station observation, etc., at the preceding time. The NWP models can capture the important triggers which can be hardly found out through observations to predict rainfall. This heavy rainfall case occurred due to the

mesoscale factors: 1) low-level convergence, 2) strong vertical wind shear, 3) coastal fronts and back-building convection bands, 4) mid-level advection of cold air and positive relative vorticity, and 5) vigorous updraft releasing potential instability (Park and Park, 2020). Therefore, it is necessary to improve the NWP model to more accurately identify these mesoscale factors.

## 4.2 Results

### 4.2.1 Combinatorial optimization of the physics schemes for QPF

The combinatorial optimization of the physics schemes in the WRF model is targeting the improved quantitative forecasting of heavy rainfall. OPT-EXP1 shows the simulated results using the optimized combination of MP, CU, and PBL schemes, focusing on strong precipitation intensity. Figure 3 depicts the evolution of generations of the $\mu$-GA, represented by fitness values, for OPT-EXP1. The $\mu$-GA reached the maximized evolution, which was the point that the best individual in each

generation converged upon the highest fitness score (here, of 4.292), at the 12nd generation. Before that, the local optimum (i.e., intermediate optimum; IMD-OPT) appeared at 4rd generation with a fitness of 2.9896. The optimized schemes of CU, MP, and PBL for this event are MSKF, NSSL-2 moment, and YSU scheme, respectively. For the IMD-OPT-EXP1, only the MP physic scheme selected as WSM 6-class is different from the global optimum. The optimum of PBL scheme is the same as CTL, and that of CU scheme is the updated scheme from CU scheme for CTL. The KF (i.e., CU scheme for CTL) is

suitable for a horizontal resolution of ~25 km, at which convective clouds can be represented explicitly. However, MSKF (i.e., the optimum of CU scheme) has been improved for use in the so-called grey zone scales (i.e., 12 to 1 km) as well as at horizontal grid spacing of 25 km (Glotfelty et al., 2019).

Figure 4 compares the ETSs of each precipitation threshold for CTL, OPT-EXP1, IMD-OPT, the best individual at 1st generation (GEN1), and REF. REF is simulated with the CU, MP, and PBL physics schemes validated in Park and Park

(2020) — Morrison scheme is chosen as MP scheme, and CU and PBL schemes are the same as CTL. The selected schemes for CTL, OPT-EXP1, IMD-OPT, GEN1, and REF are summarized in Table 4. OPT-EXP1 performs significantly better than CTL at precipitation thresholds above 20 mm, indicating a remarkable improvement in ETSs. Although REF also shows the improved forecasting skill than CTL at all precipitation thresholds, OPT-EXP1 performs better at higher precipitation intensity compared to REF. The optimization process from GEN1 to IMD-OPT, corresponding to the evolution of early

generations, shows increases in ETS at precipitation thresholds less than 130 mm, whereas IMD-OPT to OPT-EXP1 shows





further enhancements at heavy precipitation thresholds (≥40 mm) and even above 130 mm. For both observation and model output, the maximum amount of 12 hour accumulation precipitation in a grid box did not exceed 190 mm.

The spatial distribution of 12-hr accumulated precipitation for observation (RAR), OPT-EXP1, CTL, and REF are shown in Fig. 5. CTL broadly underestimated precipitation, and REF is better than CTL in terms of both quantity and rain cell
development. Rain cells located near both Sokcho and north of Gangneung were well captured in REF and OPT-EXP1 although the amount of precipitation underestimated over the north of Gangneung. OPT-EXP1 shows the improved precipitation simulation for inland area near Gangneung and Daegwallyeong, but it still underestimated. As the MCS was staying near Sokcho and north of Gangneung during the period of accumulative precipitation, light rainfall occurred over inland whereas heavy rainfall occurred over the sea along the coastal line.
We also verified the effectiveness of the optimization by the continuous statistics for CTL, REF, and OPT-EXP1 (Table 5 and Fig. 6). The scatter plot for OPT-EXP1 exhibits the best performance with the correlation coefficient (R) of 1.01, compared to CTL and REF (Fig. 6). In addition, Table 5 shows OPT-EXP1 has lower spatial mean bias and Root Mean Square Error (RMSE) of precipitation (-7.433 and 21.511) and greater Pearson's Correlation Coefficient (PCC) (0.762) than CTL (-8.696, 25.430, and 0.673, respectively). It performs better than REF as well. In conclusion, combinatorial
optimization of physics schemes has enhanced the forecast skill not only in QPF (i.e., ETS) but also in terms of both spatial distribution and continuous statistics.

### 4.2.2 Sensitivity of fitness functions based on the assessment of precipitation occurrence

For the accuracy of the deep convective precipitation system, we wonder whether it would be effective to increase the accuracy of the precipitation occurrence or to increase the accuracy of precipitation within each precipitation threshold. In
this section, we conduct the sensitivity test of accumulated precipitation time interval used in fitness function calculation to evaluate the precipitation occurrence with a precipitation threshold of 3 mm (see Table 3). The ETSs for 12-hr accumulated precipitation, calculated by using 12 hourly (OPT-EXP2), 6 hourly (OPT-EXP3), 3 hourly (OPT-EXP4), 1 hourly (OPT-EXP5) accumulated precipitation data, were evaluated at each time interval. In contrast to OPT-EXP1, precipitation thresholds for them are set as one criterion (i.e., precipitation threshold = 3 mm), so the maximum value of ETS is equal to 1.
When the accumulation time interval become shortened (e.g., an hour), the precipitation prediction must also be more accurate on a temporal scale in order to have the higher fitness. Because the fitness is computed by the average of ETSs calculated at each time interval. In other words, the shorter the accumulated time intervals, the more ETSs of predicted precipitation are evaluated. Thus, as expected, OPT-EXP2 shows the highest fitness value (i.e., 0.3482), followed by OPT-EXP3 with the fitness of 0.2862, and OPT-EXP5 with the lowest fitness of 0.2249 (Fig. 7). OPT-EXP4 performs similar as
OPT-EXP5, having the fitness of 0.2270. The selected schemes for OPT-EXP2, OPT-EXP3, OPT-EXP4, and OPT-EXP5 is shown in Table 6.

Figure 8 shows the spatial distribution of 12-hr accumulated precipitation for OPT-EXP2 to OPT-EXP5. All experiments underestimate the convective system and overestimated very light precipitation over inland area of Korean Peninsula (see





Fig. 5 and Fig. 8). From the ETS perspective, OPT-EXP2 is the best result, but OPT-EXP3 shows best simulation results in terms of the spatial distribution. Since no method is existed absolutely superior to others in precipitation evaluation methods such as ETS, CSI, POD, and continuous statics indices, several indices including spatial distribution must be examined together. Rain cells located near both Sokcho and north of Gangneung were well captured only in OPT-EXP3 and OPT-EXP4, but were still underestimated. Rainfall over the sea along the coastal line was simulated in OPT-EXP3 and OPT-EXP4 as well. On the other hand, the evaluation of the fitness at 1 hour intervals results in poor accuracy, possibly because of including the time phase error of the model.

Figure 9 depicts the scatter plot for OPT-EXP2 to OPT-EXP5. OPT-EXP3 and OPT-EXP4 lead to better agreement with observed precipitation than OPT-EXP2. OPT-EXP3 has a highest correlation (1.13) followed by OPT-EXP4. Moreover, OPT-EXP4 has the lowest RMSE and the greatest PCC of precipitation (23.952 and 0.731, respectively) (Table 7). OPT-EXP3 has the lowest spatial mean bias (-8.690). In terms of fitness, OPT-EXP2 is superior, but OPT-EXP3 and OPT-EXP4 shows the better simulations in terms of both the spatial distribution and continuous statistics.

In this section, the sensitivity of the accumulation time interval of precipitation used in fitness function calculation (i.e., ETS) to the optimization in $\mu$-GA-WRF interface system was briefly examined. In the current model performance, the best result of the optimization experiments can be obtained by using the 3 or 6 hourly accumulated precipitation as a fitness function when focusing on precipitation detection. However, compared to OPT-EXP1, both the quantitative precipitation and spatial distribution in OPT-EXP1 was much more improved than other experiments (i.e., OPT-EXP2 – OPT-EXP5). Therefore, in order to improve the simulation of deep convective systems, it is recommended to evaluation of the precipitation accuracy at various precipitation thresholds rather than assessing the accuracy of precipitation occurrence.

### 4.3 Discussions

All physics schemes including CU, MP, PBL, radiation, and surface schemes are interrelated, and a non-linear relationship among them is appeared due to the complexity of atmospheric system. Thus, in order to accurately predict precipitation, it is necessary to explore the combination of physics schemes rather than focusing only on each one. The evolutionary approach to find the optimum combination of CU, MP, and PBL schemes can provide insightful understanding of the implemented physical schemes and their interrelationships. The accuracy of precipitation of less than 30 mm in large areas has been improved by fitted CU and PBL schemes. On the other hand, the simulation accuracy of high intensity precipitation occurred in a small area was improved by MP schemes. It is because MP schemes control grid-resolving scale precipitation while CU schemes determine sub-grid scale precipitation. In other words, a realistic parameterization of cloud microphysics is crucial for the precipitation forecast in high resolution models. In addition, selected CU scheme (i.e., MSKF scheme) also has been improved for use in the so-called grey zone scales (e.g., 5 km used in this study).

However, it is difficult to insist that the order of fitting scheme is directly related to the importance of the scheme in QPF because of the non-linear relationship between precipitation and physics schemes as well as among the physics schemes. Moreover, it can be note that the combination of randomly selected schemes in the first generation approaches the optimal





solution, allowing the fitness function to converge quickly. For example, in this study, both PBL and CU scheme are fortunately selected the same as the optimum, and this combination has a higher fitness value than the other combinations. Thus, in the $\mu$-GA evolution process, the information of optimized CU and PBL scheme in the best individual was inherited

by elitism, and MP scheme of the best individual was changed to be optimized through generations. GA with a randomly selected initial population is robust in finding solutions as enough generations pass, but still the initial population affects the convergence velocity (i.e., the generation with optimal solution). As a result of sensitivity tests with different initial populations, convergence occurred after the 50th generation in one test, and MP converged earlier than CU and PBL in the other. In summary, optimized results do not depend on the initial population of the first generation, but the initial population

may affect which scheme will be optimized first.

The simulation results of Park and Park (2020) is more accurate than REF because of the different domain setting. For this case as a localized heavy rainfall, more accurate precipitation simulations can be achieved when a specific region is set as a model domain or when multiple nested domains are used. However, the selected domain in this study may be reasonable to derive a general scheme combination that accurately simulates precipitation over the Korean Peninsula. It is necessary to

derive a general set of physics schemes for accurate precipitation simulations through several case studies as a further study.

## 5 Conclusions

The uncertainties related to the subgrid-scale parameterizations significantly increase as NWP models become more complex. The accuracy of subgrid-scale parameterizations depends on both parameters in physics schemes and the choice of parameterization schemes for each corresponding physical process. In this study, we created the $\mu$-GA-WRF interface to

seek the optimal set of physics parameterization schemes in the WRF model. The GA is founded on the natural selection and evolution to search the optimum, and the $\mu$-GA is an efficient version of GA. In weather and climate studies, most GA applications have focused on optimizing the empirical parameters of NWP models to represent a real atmospheric system, while the current study attempts to find an optimal combination of parameterization schemes, a novel and challenging task. Because of the nonlinear relationship among physics schemes, it is recommended to optimize several interesting schemes

concurrently in the WRF model rather than optimizing the scheme in each physics category sequentially. GA is an appropriate optimization method in that it can handle the nonlinearity of the parameters to be optimized.

The experiments were conducted on the optimal set of CU, MP, and PBL schemes in terms of QPF for a heavy rainfall event in Korea, through the $\mu$-GA-WRF interface system. The $\mu$-GA successfully improved simulated precipitation in spite of the non-linear relationship between precipitation and physics schemes as well as among the physics schemes. The $\mu$-GA reached

its maximum evolution in the 12nd generation and led to the significant improvement in ETSs, especially at a threshold range of 20 – 180 mm. The optimized schemes of CU, MP, and PBL for this event are MSKF, NSSL-2 moment, and YSU scheme, respectively. During the evolution process, MP schemes control grid-resolving scale precipitation while CU and PBL schemes determine sub-grid scale precipitation.



We also conduct the sensitivity test of accumulated precipitation time interval used in fitness function (i.e., ETS) with
precipitation threshold of 3 mm. The best result of the optimization experiments has obtained by using the 3 or 6 hourly
accumulated precipitation. However, in order to improve the simulation of deep convective systems, it is recommend to
evaluate the accuracy of precipitation at various precipitation thresholds (i.e., precipitation thresholds = 10, 20, 30, …, 300
mm) rather than assessing the accuracy of precipitation occurrence (e.g., precipitation threshold = 3 mm).

Note that prior to model simulation unknown parameters and schemes should be fitted to the regional weather/climate to
reduce considerable uncertainties in NWP models. In addition, in terms of model development, all physics schemes need to
be explored to simulate more accurate local weather and climate systems if sufficient computer resources and time are
available. This study has demonstrated that the combinatorial optimization of physics schemes in WRF model is one of
possible solutions to enhance the forecast skill of the regional or local prediction. We also significantly reduced the number
of model simulations for optimization using GA, one of the artificial intelligence methods. Furthermore, the experiments for
combined scheme-based with parameter-based optimization are essentially required to investigate the effect of parameter
calibrations on the model sensitivity to scheme selections. As a further study, we strongly suggest to conduct comprehensive
parameter and scheme estimation to improve the model performance.

**Code and data availability**

The current version of the WRF model is available from the github website: https://github.com/wrf-model/WRF. The GA
code, used in this study, was developed by David L. Carroll and last updated on 2 April 2001. The current version of the GA
driver is available from the website: https://cuaerospace.com/products-services/genetic-algorithm/ga-drive-free-download.
The exact versions of both the WRF model and the GA driver, used to produce the results in this study, are archived on
Zenodo (https://doi.org/10.5281/zenodo.5076930), along with the input data, namelist files, and scripts to run the model and
produce the plots of all the simulations presented in this study. The NCEP FNL Operational Model Global Tropospheric
Analyses data, used for the initial and boundary conditions of the WRF model, can also be downloaded from the website of
Research Data Archive of NCAR: https://rda.ucar.edu/datasets/ds083.2/. The RAR-estimated rainfall data were obtained by
the Korea Meteorological Administration (KMA). The KMA does not provide this data set through the public service, called
"Open MET Data Portal" (https://data.kma.go.kr/resources/html/en/ncdci.html), but one can obtain the data via separate
request to the KMA.

**Author contributions**

SP contributed to conceptualization; experiments and validation design; performing experiments and validations; writing of
submitted manuscript, data acquisition and handling; software; and graphical and formal analyses. SKP contributed to



conceptualization; experiments design; supervision; funding acquisition; and project administration. All authors have read and agreed to the published version of the manuscript.

## Competing interests

The authors declare that they have no conflict of interest.

## Acknowledgements

This work was funded by the Korea Meteorological Administration Research and Development Program under Grant KMIPA2018-06710. Also, this work is supported by Basic Science Research Program through the National Research Foundation of Korea (NRF) funded by the Ministry of Education (2018R1A6A1A08025520).

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





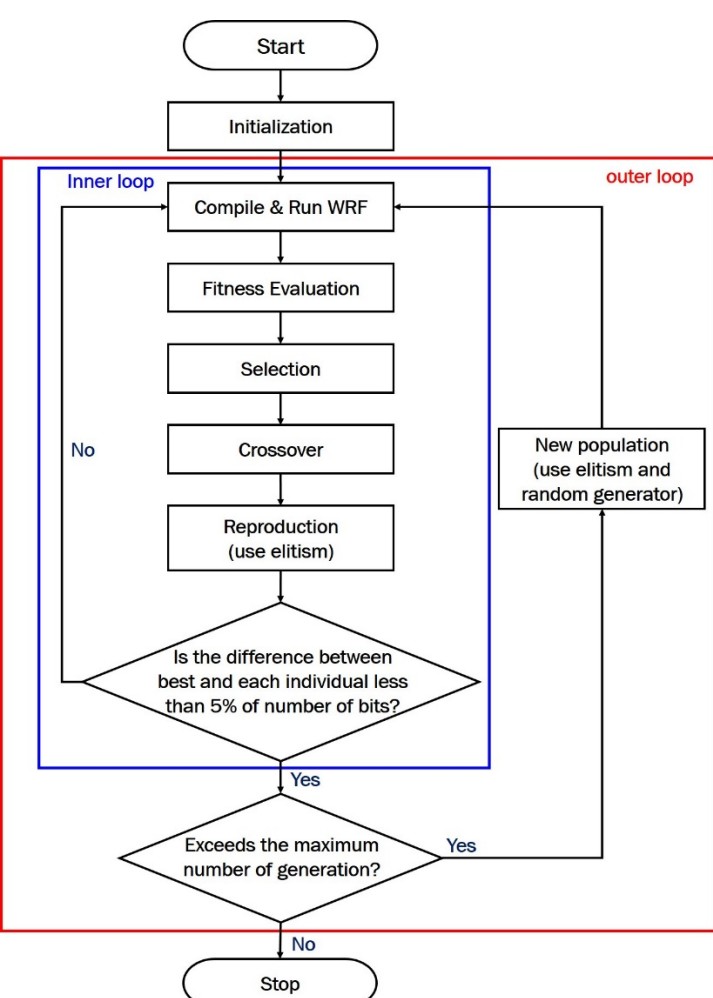

Figure 1: The flowchart of the **μ-GA-WRF** interface system.





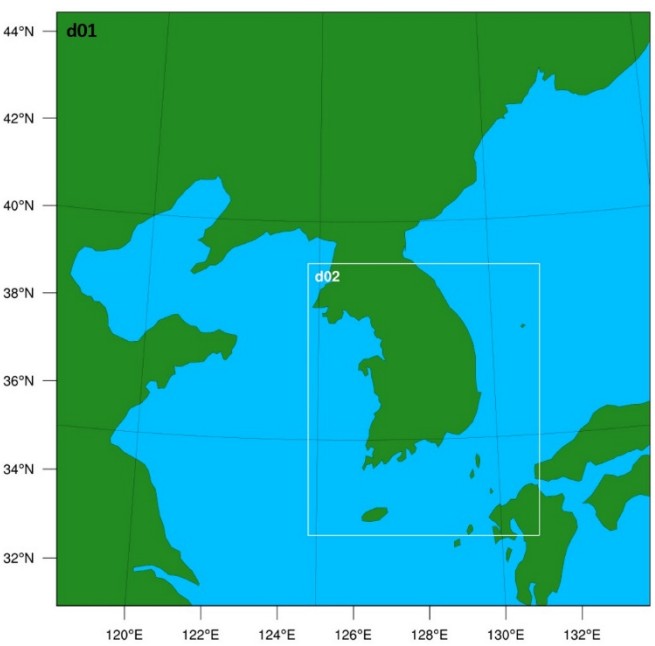

**Figure 2: WRF nested domains — Domain 1 (d01) and Domain 2 (d02). d01 is centered at 38 °N and 126 °E in the Lambert conformal conic projection.**

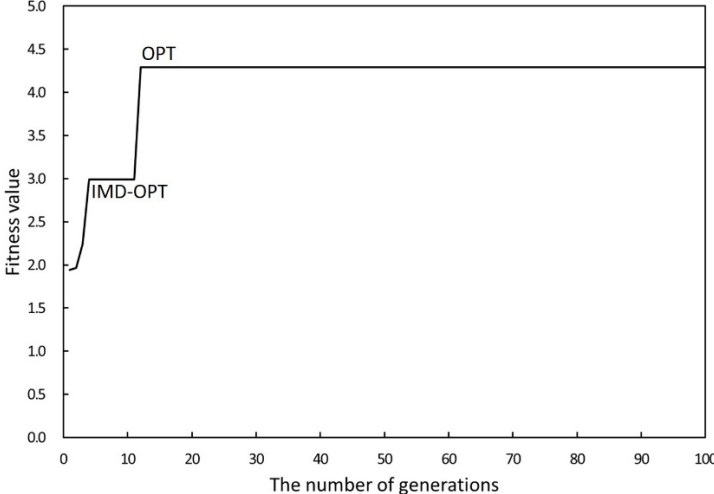

**Figure 3: Evolution of generations in the optimization process, leading to changes in fitness values. OPT and IMD-OPT represent**
**the final optimized phase and one of intermediate optimized phases, respectively.**



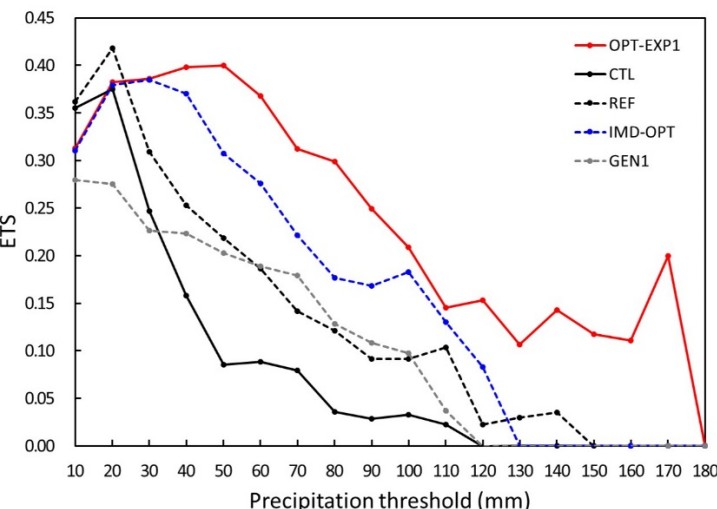

**Figure 4꞉ ETS for 12-hr accumulated precipitation of each precipitation threshold for CTL (black line), OPT-EXP1 (red line), IMD-OPT (dashed blue line), GEN1 (dashed gray line), and REF (dashed black line).**

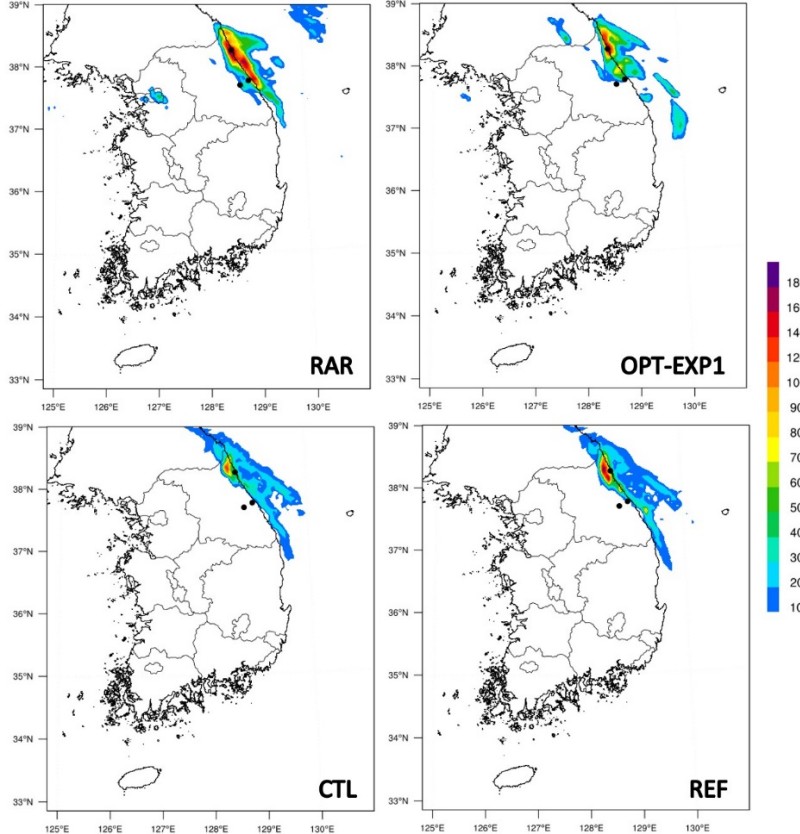


**Figure 5꞉ The spatial distribution of 12-hr accumulated precipitation of RAR, CTL, OPT-EXP1, and REF. Black dots depict locations of weather stations: Sokcho (S; 38.25 ºN, 128.56 ºE; 18.06 m), Gangneung (G; 37.75 ºN, 128.89 ºE; 26.04 m), and Daegwallyeong (D; 37.68 ºN, 128.86 ºE; 772.57 m).**





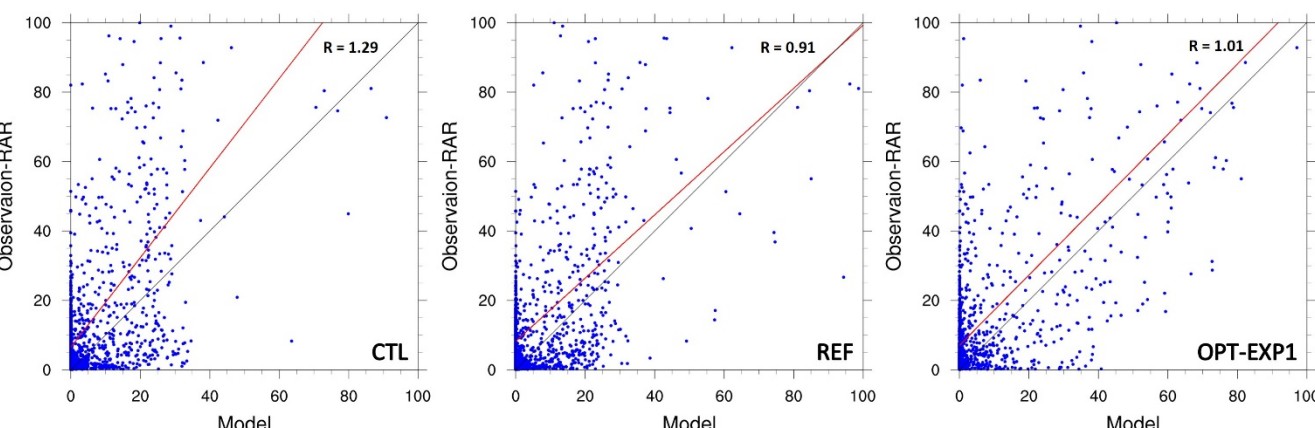

**Figure 6: Scatter plot of CTL, REF, and OPT-EXP1.**

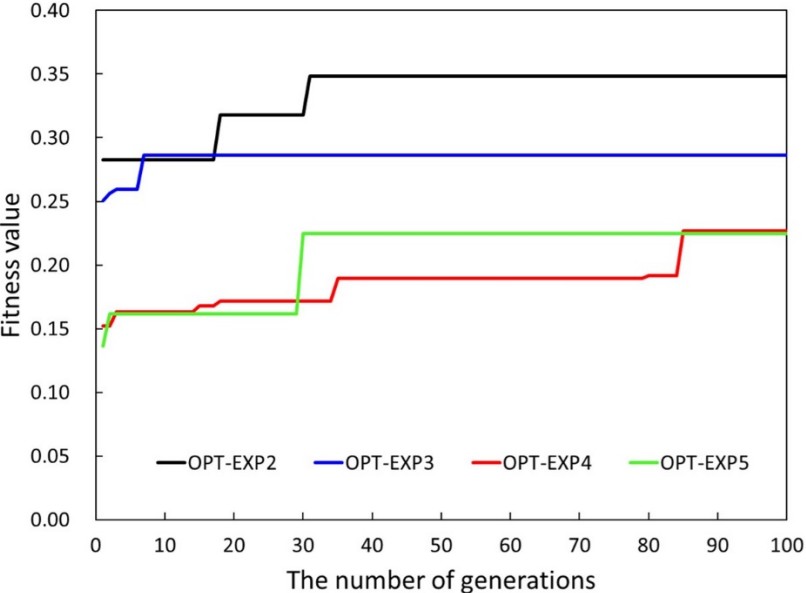

**Figure 7: Same as Fig. 3 but for OPT-EXP2 to OPT-EXP5.**



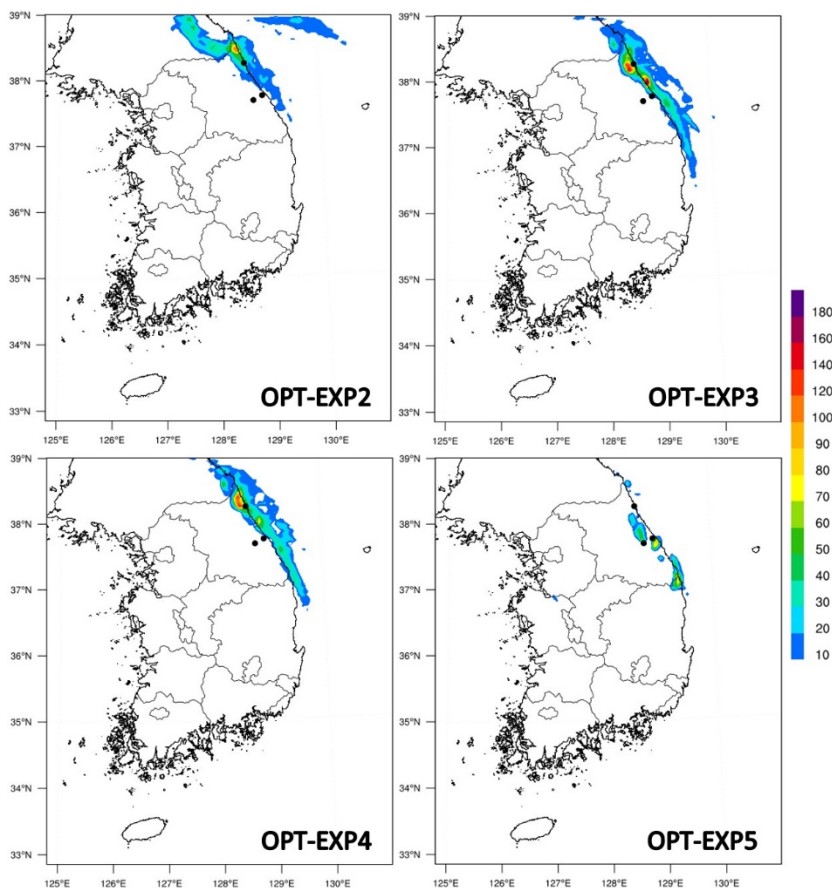


**Figure 8: Same as Fig. 4 but for OPT-EXP2 to OPT-EXP5.**





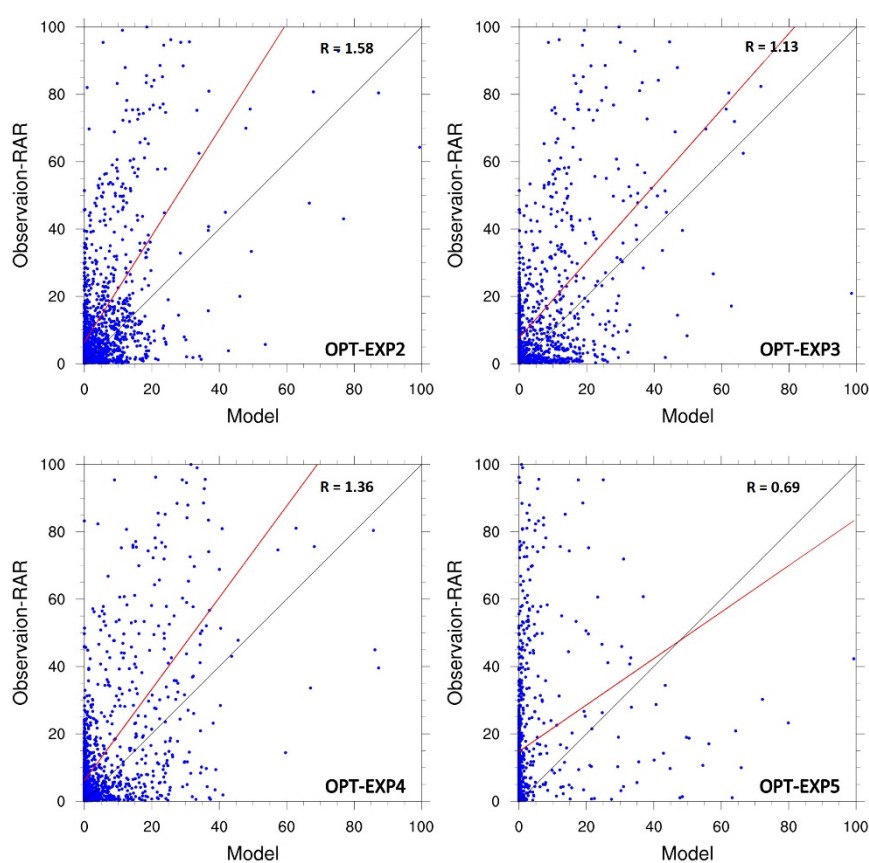

**Figure 9: Scatter plot of OPT-EXP2 to OPT-EXP5.**


**Table 1. A 2 × 2 contingency table.**

|                  |     | Observed events |                  |
| ---------------- | --- | --------------- | ---------------- |
|                  |     | Yes             | No               |
| **Forecast events** | Yes | hits            | False alarms     |
|                  | No  | misses          | Correct negatives |




**Table 2. Summary of CU, MP, and PBL physics schemes used in the *μ*-GA-WRF interface system for the optimization.**

| CU (14) | MP (16) | PBL (12) |
|---|---|---|
| no cumulus | Kessler (Kessler, 1969) | YSU (Hong et al., 2006) |
| KF (Kain, 2004) | Purdue Lin (Chen and Sun, 2002) | MYJ (Janjic, 1994) |
| Betts-Miller-Janjic (Janjic, 1994) | WRF Single-Moment (WSM) 5-class (Hong et al., 2004) | QNSE (Sukoriansky et al., 2005) |
| Grell-Freitas ensemble (Grell and Freitas, 2014) | Eta microphysics (NOAA, 2001) | MYNN 2.5 (Nakanishi and Niino, 2006, 2009; Olson et al., 2019) |
| Old Simplified Arakawa-Schubert (SAS) (Pan and Wu, 1995) | WSM 6-class (Hong and Lim, 2006) | MYNN 3 (Nakanishi and Niino, 2006, 2009; Olson et al., 2019) |
| Grell 3D ensemble (Grell, 1993; Grell and Devenyi, 2002) | Goddard (Tao et al., 1989; Tao et al., 2016) | Asymmetric Convective Model (ACM) 2 (Pleim, 2007) |
| Tiedtke (Tiedtke, 1989; Zhang et al., 2011) | Thompson (Thompson et al., 2008) | Bougeault and Lacarrere (BouLac) (Bougeault and Lacarrère, 1989) |
| KF-Cumulus Potential (KFCP) (Berg et al., 2013) | Milbrandt-Yau 2-moment (Milbrandt and Yau, 2005a; 2005b) | University of Washington Boundary Layer (Bretherton and Park, 2009) |
| Multi-Scale KF (MSKF) (Zheng et al, 2016; Glotfelty et al., 2019) | Morrison 2-moment (Morrison et al., 2009) | TEMF (Angevine et al., 2010) |
| KIAPS SAS (Kwon and Hong, 2017) | CAM V5.1 2-moment (Eaton, 2011) | Shin-Hong Scale-aware (Shin and Hong, 2015) |
| NewTiedtke scheme (Han and Pan, 2011) | Stony Brook University (SBU) (Lin and Colle, 2011) | Grenier-Bretherton-McCaa (GBM) (Grenier and Bretherton, 2001) |
| Previous NEW GFS simplified Arakawa-Schubert scheme from YSU (Han and Pan, 2011) | WDM 5-class (Lim and Hong, 2010) | Medium Range Forecast (MRF) (Hong and Pan, 1996) |
| Grell-Devenyi ensemble (Grell and Devenyi, 2002) | WDM 6-class (Lim and Hong, 2010) | |
| Old KF (Kain and Fritsch, 1990) | NSSL 2-moment (Mansell et al., 2010) | |
| | NSSL 1-moment (Mansell et al., 2010) | |
| | P3 2-moment (Morrison and Milbrandt, 2015) | |





**Table 3. The summary of the experiments. In fitness function, _i_ and _t_ represents precipitation threshold and the number of accumulated time instants, respectively. N is the total number of accumulated time instants.**

| | Accumulated time (hr) | Precipitation threshold (mm) | Fitness function |
|---|---|---|---|
| **OPT-EXP1** | 12 | 10, 20, 30,…, 300 | $Fitness = \sum_i ETS_i^{t=1}$, <br> $i = 10, 20, 30, \cdots, 300$ |
| **OPT-EXP2** | 12 | 3 | $Fitness = ETS_{i=3}^{t=1}$ |
| **OPT-EXP3** | 6 | 3 | $Fitness = \frac{1}{N} \sum_t ETS_{i=3}^t$, <br> $t = 1, 2$ |
| **OPT-EXP4** | 3 | 3 | $Fitness = \frac{1}{N} \sum_t ETS_{i=3}^t$, <br> $t = 1, 2, 3, 4$ |
| **OPT-EXP5** | 1 | 3 | $Fitness = \frac{1}{N} \sum_t ETS_{i=3}^t$, <br> $t = 1, 2, 3, \cdots, 12$ |


**Table 4. The summary of selected schemes for CTL, OPT-EXP1, IMD-OPT, GEN1, and REF.**

| | CTL | OPT-EXP1 | IMD-OPT | GEN1 | REF |
|---|---|---|---|---|---|
| CU | KF | MSKF | MSKF | MSKF | MSKF |
| MP | WDM 6 | NSSL 2-moment | WSM 6 | Purdue Lin | Morrison |
| PBL | YSU | YSU | YSU | YSU | YSU |

**Table 5. Continuous statistics of CTL, REF, and OPT-EXP1.**

| | CTL | REF | OPT-EXP1 |
|---|---|---|---|
| Bias | -8.70 | -7.43 | **-7.17** |
| RMSE | 25.43 | 24.94 | **21.51** |
| PCC | 0.67 | 0.65 | **0.76** |





**Table 6. The selected schemes for OPT-EXP2, OPT-EXP3, OPT-EXP4, and OPT-EXP5.**

|  | OPT-EXP2 | OPT-EXP3 | OPT-EXP4 | OPT-EXP5 |
|---|---|---|---|---|
| CU | KFCP | KF | KF | KFCP |
| MP | CAM V5.1 2-moment | P3 2-moment | WSM 6-class | WDM 6-class |
| PBL | MRF | MYNN 2.5 | MYNN 3 | MYNN 2.5 |

**Table 7. Continuous statistics of OPT-EXP2 to OPT-EXP5.**

|  | OPT-EXP2 | OPT-EXP3 | OPT-EXP4 | OPT-EXP5 |
|---|---|---|---|---|
| Bias | -10.01 | -8.69 | -8.76 | -14.11 |
| RMSE | 27.34 | 24.03 | 23.95 | 32.17 |
| PCC | 0.63 | 0.70 | 0.73 | 0.26 |
