# Peer review of "A micro-genetic algorithm (GA v1.7.1a) for combinatorial optimization of physics parameterizations in Weather Research and Forecasting model (v4.0.3) for quantitative precipitation forecast in Korea"

_Geoscientific Model Development, 2021_

## Author Comment (AC2)

Reply to the Comments by Referee #1 for Manuscript gmd-2021-143 "A micro-genetic algorithm for combinatorial optimization of physics parameterizations in Weather Research and Forecasting model for quantitative precipitation forecast in Korea"

**General Comment:**

*The paper "A micro-genetic algorithm for combinatorial optimization of physics parameterizations in Weather Research and Forecasting model for quantitative precipitation forecast in Korea", by S. Park and S. K. Park, analyzes the importance of optimizing the selection of the parameterization physics schemes, and their parameters, in improving the prediction of extreme rainfall at the mesoscale in Korea.*

*The paper is generally well written and carefully describes the performed experiments. The paper appears scientifically sound and the degree of novelty is high. However there are some small points to be improved before publication, in my opinion.*

⇒ We appreciate the positive and valuable comments by the referee, which helped us improve the quality of the manuscript. We have revised the manuscript following the referee's specific comments, including some corrections and suggestions. An item-by-item response to the comments is provided below.

**Specific comments:**

i) *The paper is based on the comparison between several simulations performed with optimized WRF and a single meteorological event. However, the event is only shortly described (section 4.1). There are two references: KMA (2018) shows a page in Korean in which, for a foreigner, is not easy to understand (and in any case does not contain a meteorological description of the event). Park and Park (2020) is another paper. In my opinion, a paper should be self-consistent and contain all indication to understand the problem; reference to other papers could be used only for details. For instance, Sokcho was the location showing the largest precipitation? During the 27 hours of rainfall there, there were some time periods in which rainfall was more intense?*

⇒ We appreciate the referee pointing this out. We have revised the manuscript by adding the detailed description of the event as the followings:

3.3. Experimental design
… During the 12-hr period from 1200 UTC 5 to 0000 UTC 6 August 2018, **including the first and second periods of intense rainfall (see Section 4.1),** precipitation was evaluated by fitness functions…

4.1 Case description
… For the period from 1100 UTC 5 to 1400 UTC 6 August, 294.5 mm of precipitation was recorded at Sokcho**: the first intense rainfall continued for 4 hrs (13:00 UTC – 17:00 UTC 5 August) with the maximum precipitation rate of 35.3 mm/h and total rainfall amount of 83.5 mm, whereas the second intense rainfall (17:00 UTC 5 – 0:00 UTC 6 August) recorded the maximum precipitation rate of 54.9 mm/h and total rainfall amount of 192 mm, due to the**

**quasi-stationary MCS.** To predict more accurately, forecasters essentially need the mesoscale information from NWPs …

ii)    *Maps in figures 5 and 8 show the precipitation in the second domain for all simulations. Since the portion of Korean territory (and adjacent sea) interested by the event is much smaller than the domain, I suggest to zoom on the portion of territory interested by the rainfall (about a quarter of the domain), in order to highlight the details. The other part of the territory is not important in this sense, since there was no any precipitation. In this way, it could be possible to better appreciate the structure of the precipitation area.*

⇒    Figures 5 and 8 have been modified to zoom in on the area of interest, showing the major precipitation systems in more detail.

iii)   *it is very clear that this result appears quite interesting, since it shows the importance of using an accurate choice of the physics parameterizations schemes. However, this result could be dependent on the case study. In this sense, if possible, it could be interesting for the reader to add, in the discussions, a sentence in which there is an attempt to understand why some schemes perform better than others, from a physical point of view.*

⇒    We appreciate the referee pointing this out. We explained why the selected schemes have better forecast performance than the others in the revised manuscript as:

**A typical cumulus convection can be represented by the CU schemes at horizontal grid spacing of about 25 km. However, the selected CU scheme (i.e., MSKF) has been improved for use in the so-called grey zone scales (e.g., 5 km used in this study); thus, it can outperform the other CU schemes. On the other hand, the KFCP scheme that is modified to better account for the presence of shallow clouds was selected for OPT-EXP2 and OPT-EXP5 possibly because their fitness functions were focused on the precipitation occurrence. Note that the single-moment MP schemes predict the mixing ratio of hydrometeors by representing the hydrometeor size while the double-moment schemes also predict number concentrations of hydrometeors. Thus, the double-moment schemes (e.g., NSSL 2-moment, WDM 6, Morrison) can produce a reasonable concentration of large droplets for a heavy precipitation system, compared to the single-moment schemes (Lim and Hong, 2010). In addition, the YSU scheme, representing the PBL process, more accurately simulates a deeper vertical mixing in the thermally-induced free convection regime covering multiple vertical levels (Hong et al., 2006), thus being superior to the other schemes for the simulated precipitation.**

Lim, K.-S.S. and Hong, S.-Y.: Development of an effective double-moment cloud microphysics scheme with prognostic cloud condensation nuclei (CCN) for weather and climate models, Mon. Wea. Rev., 138, 1587–1612, doi:10.1175/2009MWR2968.1, 2010.
Hong, S.-Y., Noh, Y., and Dudhia, J.:   A new vertical diffusion package with an explicit treatment of entrainment processes, Mon. Wea. Rev., 134, 2318–2341, doi:10.1175/MWR3199.1, 2006.

*Language: I am not a native speaker, so my opinion on the language, very good, could be biased. There are some minor typos to be corrected (e.g .: gird -> grid, Ninno -> Niño).*

⇒    Corrected. We appreciate the reviewer for checking out these typos.

---

## Author Comment (AC3)

**Reply to the Comments by Referee #2 for Manuscript gmd-2021-143 "A micro-genetic algorithm for combinatorial optimization of physics parameterizations in Weather Research and Forecasting model for quantitative precipitation forecast in Korea"**

**General Comment:**

This study used the micro-genetic algorithm and performed a combinatorial optimization of cumulus, microphysics, and planetary boundary layer schemes from the WRF model to improve the quantitative precipitation forecast of a heavy rainfall event in Korea. Five different fitness functions are chosen in terms of Equitable Threat Score to obtain the corresponding best combination of physics schemes for each fitness function. The methodology adopted in this study is explained in great detail, and this method could be very useful to obtain the best combination of physics parameterization schemes with a minimal number of model runs instead of performing an exhaustive search. The manuscript is well written in general, and the results are presented with clarity. Having said this, I have a few comments on the manuscript.

 $\Rightarrow$  We appreciate the referee's positive and detailed review. We have revised the manuscript in accordance to specific comments provided by the referee, which significantly enhance the quality of the manuscript. An item-by-item response to the comments is provided below.

**Specific comments:**

- 1. The main concern with this work is that the results obtained from this study are very specific to the heavy rainfall event that was considered. Recent studies on parameter estimation, such as Duan et al. (2017) and Di et al. (2018), have performed simulations over multiple heavy rainfall events to obtain a set of optimal parameters to improve the precipitation prediction. Since only a single event was considered in this study, the robustness of the results obtained could not be ascertained. It would be good if the authors could repeat this exercise for a few more heavy rainfall events and present a comprehensive result. The authors should at least try to check how the optimum schemes (OPT-EXP1) are performing for other heavy rainfall events compared to REF and CTL.
- ⇒ We appreciate the referee pointing this out. As recommended by the reviewer, we have conducted new simulations using the optimum schemes (i.e., OPT-EXP1) for other heavy rainfall events to verify the performance compared to REF and CTL.
- ⇒ Figure R1 below for another heavy rainfall event shows the spatial distribution of 24-hr accumulated precipitation from 0100 UTC 3 to 0000 UTC 4 July 2014 for RAR, OPT-EXP1, CTL, and REF. All simulations underestimated the precipitation intensity along the coastline, whereas overestimated over the inland. However, OPT-EXP1 captured rain cells (i.e., C1 and C2) well compared to REF and CTL.

Figure R2, for the other heavy rainfall event, shows the spatial distribution of the 24-hr accumulated precipitation from 0100 UTC 25 to 0000 UTC 26 August 2015. Generally, REF shows the best performance rather than OPT-EXP1 and CTL, though it overestimated the precipitation compared to RAR. However, the rain cells were well captured along the coastal line for all

simulations. The OPT-EXP1 predicted slightly better the region of precipitation than REF. Although OPT-EXP1 overestimated precipitation over the sea on the east side, it would not have significant impact on the accuracy of prediction over the inland.

⇒ We agree with the reviewer's opinion that the robustness of the optimized physics schemes cannot be confirmed by a specific heavy rainfall event. The main focus of this study was to emphasize the importance of the optimization of physics scheme sets as well as physical parameters, and to introduce the automatic optimization method (i.e., the  $\mu$ -GA-WRF interface system) as a potential tool to improve precipitation forecast. The scope of this study did not include finding the robust optimal physics scheme sets for forecasting well all the precipitation cases in Korea: it is because there are several different categories of heavy rainfall cases that depends on different mechanisms of initiation and development in Korea. Therefore, we have not included the verification results from the other events in the revised manuscript. However, we have added the discussions on the limitation and the robustness of the optimum scheme set, and the relevant references suggested by the reviewer — Duan et al. (2017) and Di et al. (2018) — in the revised manuscript as follows:

In recent studies, optimization experiments for parameter estimation for multiple heavy rainfall events have been conducted to obtain a set of optimal parameters to improve the precipitation prediction (e.g., Duan et al., 2017; Di et al., 2018). We address that the optimized scheme set obtained in this study is specific to the selected rainfall case or at the best to the rainfall systems that occur under similar synoptic and geographical environment; thus, it is not robust to all the precipitation cases in Korea, which depend on different mechanisms of initiation and development. As a future study, we plan to perform the combinatorial optimization of physical parameterization schemes for several heavy rainfall cases under the same category in terms of location and synoptic environment, expecting to find an optimal scheme set robust to the heavy rainfall systems in that category.

- 2. In Figures 6 and 9, the Correlation coefficient, R, was mentioned in the scatter plots. But some values of R were more than 1. As the values of the correlation coefficient must lie between -1 and +1, the authors should check the calculation of the correlation coefficient
- ⇒ We appreciate the reviewer's comment. We are sorry to cause a confusion. In fact, *R* is the 'regression coefficient' that represents the slope in the scatter plot, not the 'correlation coefficient'. We have changed the word 'correlation coefficient' to 'regression coefficient' in explaining the scatter plots in the revised manuscript.
- 3. In Fig.1, The positions of Yes and No seem to have been swapped in the outer loop. As mentioned in Section 3.2, if the maximum number of generations is exceeded, the algorithm should stop.
- $\Rightarrow$  Corrected. We appreciate the reviewer pointing out this.
- 4. *Line 194, "…referred to as OTP." I think the authors meant OPT.*
- $\Rightarrow$  Corrected.

5. The language of the manuscript needs to be improved. There are some small spelling and grammatical mistakes in some places. Some of them are: Line 196; "grid" is written as "gird", Lines 235 and 345; "12th" is written as "12nd", Line 236; "4th" is written as "4rd", Line 238; "physics" is written as "physic"